# Fault-Tolerant Thrust Allocation with Thruster Dynamics for a Twin-Waterjet Propelled Vessel

**Zijing Xu** [1,*], **Roberto Galeazzi** [2] and **Jingqi Yuan** [1]

1   Key Laboratory of System Control & Information Processing, Ministry of Education of China, Department of Automation, Shanghai Jiao Tong University, Shanghai 200240, China
2   Department of Electrical and Photonics Engineering, Technical University of Denmark, 2800 Kongens Lyngby, Denmark
*   Correspondence: zijing_xu@sjtu.edu.cn

**Abstract:** The availability of the propulsion system is of primary importance to ensure safe and stable operations of marine crafts, both during transit and station keeping. Diminished propulsion efficiency could impair the ability of a vessel to maintain speed and course and possibly lead to a drifting craft. The waterjet's propulsion efficiency is affected by several factors such as cavitation, erosion, vibration and noise emission. This paper addresses the design of a fault-tolerant thrust allocation algorithm able to maintain the seaworthiness of a twin-waterjet marine craft in the presence of a severe power loss in one of the waterjets. The proposed solution combines a load torque estimator with an optimization routine that accounts for the power limits when a waterjet is subject to a power loss. This prevents faults from quickly escalating into a complete failure of the waterjet due to excessive power demands. Two simulated case studies including zig-zag path following and sideways movements are presented to demonstrate the effectiveness of the fault tolerant control thrust allocation strategy.

**Keywords:** fault tolerant control; load torque estimator; thrust allocation; path following; waterjet

## 1. Introduction

Waterjets are commonly utilized in the propulsion system of high-performance vessels [1–3] due to their ability to direct the thrust vector in any radial direction, allowing for a variety of difficult manoeuvres such as astern operation and sideways movement [4]. However, waterjets also have some disadvantages due to cavitation, impeller erosion, pump hull vibration and noise emission, which may result in a performance drop [5,6]. Furthermore, the waterjets could absorb gravel if the vessel operates in shallow waters; this, in turn, will lead to wear and tear in the pump impeller [7] with consequent degraded propulsion and steering capabilities.

The vessel studied in this paper (shown in Figure 1) is powered by a twin-waterjet propulsion system, where each waterjet is equipped with an independent steering device, a reverse duct and a diesel engine. Therefore, the waterjet thrust is determined using three variables: engine speed, steering angle and reversing angle. This kind of waterjet propelled vessel may be considered an over-actuated system, where it is possible to utilize the waterjets in a near optimal manner with efficient thrust allocation [8]. Ghassemi and Forouzan [9] proposed a combined practical approach and numerical method to design the waterjet propulsion system for marine vehicles. Ellenrieder [10] achieved free running manoeuvring trials of an unmanned surface vehicle propelled by the twin-waterjet system to investigate the effects of cross flow at the inlet of the waterjets. Xu et al. [11] developed a motion control model with three degrees of freedom for twin-waterjet propelled vessels and standard turning tests, and zig-zag tests are simulated to illustrate the good maneuverability of this kind of vessel. In comparison to the thrust allocation method used in some propeller systems, such as vessels and platforms with dynamic positioning, one of the challenges is to calculate the generated thrust properly under the combined action of engine speed,

steering angle and reversing angle [12]. Furthermore, some faults, e.g., engine malfunction or hydraulic system fault, may be experienced during the navigation [13,14], which would determine the reduced performance of the waterjet propelled vessel.

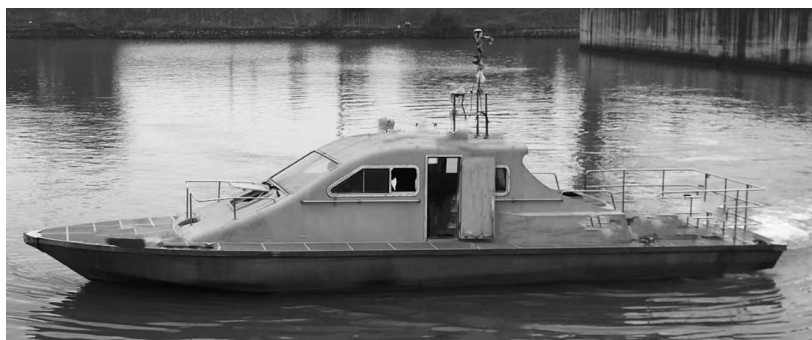

**Figure 1.** The considered twin-waterjet propelled vessel in the study.

Some recent studies have focused on developing FTC (fault tolerant control) strategies for marine propulsion systems, including limited efforts devoted to waterjet propelled vessels. Baldini et al. [15,16] presented a three-layer FTC architecture for a catamaran propelled by two azimuth thrusters, addressing the reference generation, the speed tracking control and the control allocation in the presence of faults. Omerdic and Roberts [17] designed a weighing matrix update method to compensate the partial thruster fault of an open frame ROV. Similarly, Lv et al. [18] introduced a fault tolerant control method considering the priority of thruster for an autonomous underwater vehicle. Rebhi and Nejim [19] proposed an active FTC strategy dealing with sensor and thruster faults for a twin-waterjet propelled vessel and a second order sliding mode observer is designed to reconstruct the thruster faults.

The control allocation of overactuated marine vessels propelled by thrusters has been investigated [20,21]; however, waterjet propelled vessels have been investigated to a much less extent. Ferrari et al. [22] presented an automatic berthing simulation of a waterjet catamaran. However, the considered vessel was not equipped with astern deflectors, which made the vessel an under-actuated system, thereby neglecting the control allocation issue. With a rising number of redundantly actuated waterjet thrusters for high-tech vessels, there is an urgent need to address the waterjet thrust allocation problem.

The block diagram of the FTC system developed in this study is depicted in Figure 2, where the two light green boxes represent the critical high-level and low-level control loops. The high-level control loop generates the set-points to the actuators present in the low-level control loop. The novelty resides in the integrated estimation-reconfiguration subsystem that combines a torque load observer with an adaptive thrust allocation and a reconfiguration of the heading speed to redistribute the demands to the propulsion system from the faulty to the healthy waterjet. The highlights of this investigation are generalized as the developed load torque observer for waterjet power loss fault monitoring based on the dynamical model of the thrusters and the developed fault-tolerant thrust allocation technique, which consists of the weighting adjustment within the thrust allocation optimizer and the heading speed reconfiguration by means of the estimated performance factor.

The remainder of the paper is structured as follows: Section 2 presents the dynamic modelling of the waterjet propulsion system; Section 3 introduces the fault-tolerant thrust allocation system that combines the torque load observer and the thrust optimization; Section 4 assesses the proposed FTC strategy through simulation studies of zig-zag path following and sideways movement; Section 5 presents the conclusions.

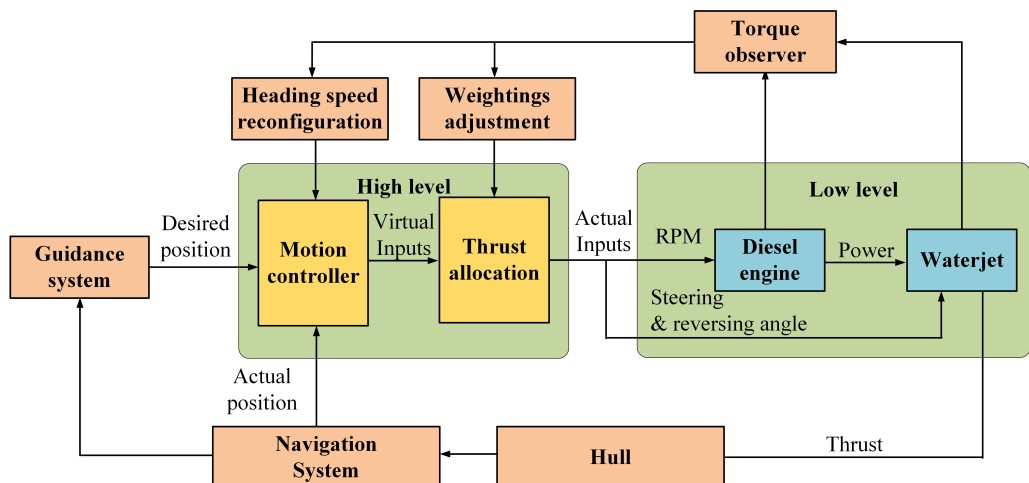

**Figure 2.** Overall control system structure of the twin-waterjet vessel.

## 2. Thruster Dynamic Modeling

### 2.1. Dynamical Model of the Diesel Engine Shaft Speed

The two equipped diesel engines produce the power for the twin-waterjet vessel according to the reference engine speed. The diesel engine considered in this article is an in-line, four-stroke, water-cooled and direct injection engine. Its specifications are listed in Table 1.

**Table 1.** Specifications of the diesel engine considered in this study.

| Description | Specification |
| --- | --- |
| Cylinder number | 6 |
| Cylinder bore | 114 mm |
| Piston stroke | 135 mm |
| Engine displacement | 8.27 L |
| Compression ratio | 16:1 |
| Engine rated speed | 2500 rpm |
| Rated power output | $450 \times (1 \pm 5\%)$ kW |
| Engine type | In-line |
| Fuel injection | Direct injection |
| Aspiration | Naturally aspirated |
| Cooling system | Water-cooled |

The nonlinear dynamical model presented in [23] is adopted to describe how the diesel engine produces the mechanical power used by the waterjet. Such a model provides the basis for the design of the waterjet load torque estimator.

The rate of the change of the engine speed $n$ (rpm) is given by the torque balance among the indicated torque $Q_i$ (N·m), the friction moment $Q_f$ (N·m) and the waterjet load torque $Q_w$ (N·m), i.e.,

$$J\frac{\pi}{30}\dot{n} = Q_i - Q_f - Q_w \tag{1}$$

where $J$ is the rotational inertia of the engine shaft. The indicated torque $Q_i$ is calculated as

$$Q_i = q_{mf}H_L\eta_i\frac{30}{\pi n} \tag{2}$$

where $q_{mf}$ (kg/s) is the fuel injection flow, $H_L$ (MJ/kg) is the constant of caloric value and $\eta_i$ is the thermal efficiency. Further, the fuel injection flow $q_{mf}$ is proportional to the engine speed $n$ [24,25] according to

$$q_{mf} = k_1 n u \tag{3}$$

where $k_1$ is a constant and $u$ is the throttle command. The friction moment is defined as follows [26]:

$$Q_f = \frac{1000 P_f V}{4\pi} \tag{4}$$

where $P_f$ (N) is the average friction force and $V$ (m$^3$) is the diesel cylinder volume. According to the motoring test for a direct-injection diesel [27], the average friction force is a function of engine speed of the form

$$P_f = 75 + \frac{48n}{1000} + 0.45 C_m^2 \tag{5}$$

where $C_m$ (m/s) is the piston's mean speed.

Combining Equations (1)–(5), the relationship between the engine speed $n$ and the throttle command $u$ reads

$$\dot{n} = \theta_1 u + \theta_2 n + \theta_3 Q_w + \theta_4 \tag{6}$$

where $\theta_j$, $j = 1, \ldots, 4$ are the model parameters given by

$$
\begin{aligned}
\theta_1 &= \frac{30}{J\pi} H_L \eta_i \frac{30}{\pi} k_1 \\
\theta_2 &= \frac{30}{J\pi} \frac{12V}{\pi} \\
\theta_3 &= -\frac{30}{J\pi} \\
\theta_4 &= -\frac{30}{J\pi} \frac{1000V}{4\pi} (75 + 0.45 C_m^2)
\end{aligned}
\tag{7}
$$

*2.2. Model of the Waterjet Thrust*

The conservation of fluid momentum [3] allows one to describe the relation between produced thrust and nozzle outlet velocity

$$T = \rho A_j v_j (v_j - \alpha v_s) \tag{8}$$

where $T$ (N) is the waterjet thrust, $\rho$ (kg/m$^3$) is density of water, $A_j$ (m$^2$) is the area of the outlet nozzle, $v_j$ (m/s) is the nozzle outlet velocity, $\alpha$ is the utilization coefficient affected by the boundary layer and $v_s$ (m/s) is the inflow velocity that is approximately equal to the vessel forward velocity.

The nozzle outlet velocity $v_j$ can be computed by conservation of kinetic fluid energy [28]:

$$H = \frac{v_j^2}{2g} + h_s - \beta \frac{v_s^2}{2g} \tag{9}$$

where $H$ (m) is the head in the operation, $\beta$ is the kinetic energy utilization coefficient, $g$ (m/s$^2$) is the gravity constant and $h_s$ is the total loss that accounts for the inlet loss $h_{in}$, outlet loss $h_{out}$, duct loss $h_d$ and head loss $h_h$. The inlet loss $h_{in}$ is calculated as:

$$h_{in} = \frac{e_{in} v_0^2}{2g} \tag{10}$$

where $e_{in}$ is the inlet loss coefficient, $v_0 = \frac{Q}{A_{in}}$, $A_{in}$ (m$^2$) is the inlet area and $Q$ (m$^3$/s) is the flow through the nozzle. The outlet loss $h_{out}$ is calculated as:

$$h_{out} = \frac{e_{out} v_j^2}{2g} \tag{11}$$

where $e_{out}$ is the outlet loss coefficient. The duct loss $h_d$ is calculated as:

$$h_d = \frac{e_d v_d^2}{2g} \tag{12}$$

where $e_d$ is the duct loss coefficient, $v_d = \frac{Q}{A_d}$, $A_d$ (m$^2$) is the duct area. The head loss $h_h$ is calculated as:

$$h_h = \frac{e_h v_j^2}{2g} \tag{13}$$

where $e_h$ is the head loss coefficient related to the potential energy loss. The increment of head energy may be described as:

$$\rho g Q H = \eta_P P_w \tag{14}$$

where $P_w$ (kW) is the nominal waterjet power and $\eta_P$ is the thrust power efficiency.

Substituting Equation (9) into Equation (14) yields:

$$P_w = \rho g A_j v_j \left( \frac{v_j^2}{2g} + h_s - \beta \frac{v_s^2}{2g} \right) / \eta_P \tag{15}$$

In an abnormal operational scenario, e.g., due to the possible damage of the impeller or the occurrence of severe cavitation, the total loss $h_s$ may increase and the power efficiency may decrease [29,30]. This will result in a reduction of the flow velocity $v_j$ with consequent reduction in thrust. If no FTC strategy is applied, the increased mechanical wear and tear will lead to deterioration of the operation.

Furthermore, the nominal waterjet power $P_w$ produced from the diesel engine is defined as [31]:

$$P_w = C \left( \frac{n}{1000} \right)^3 \tag{16}$$

where $C$ is the constant manufacturer's impeller kW absorbed at 1000 rpm.

By combining Equations (15) and (16), the velocity of the waterjet can be computed by means of the following equations

$$v_j = \left( -\frac{q}{2} - \left( \left( \frac{q}{2} \right)^2 + \left( \frac{p}{3} \right)^3 \right)^{1/2} \right)^{1/3} + \left( -\frac{q}{2} + \left( \left( \frac{q}{2} \right)^2 + \left( \frac{p}{3} \right)^3 \right)^{1/2} \right)^{1/3} \tag{17}$$

where $p$ and $q$ are the intermediate variables of the form:

$$p = g \left( h_h - \beta \frac{v_s^2}{2g} \right) / \left[ \frac{1}{2} + \frac{e_{in}}{2} \left( \frac{A_j}{A_{in}} \right)^2 + \frac{e_{out}}{2} + \frac{e_d}{2} \left( \frac{A_j}{A_d} \right)^2 \right] \tag{18}$$

$$q = -C \left( \frac{n}{1000} \right)^3 \eta_P / \left( \rho A_j \left[ \frac{1}{2} + \frac{e_{in}}{2} \left( \frac{A_j}{A_{in}} \right)^2 + \frac{e_{out}}{2} + \frac{e_d}{2} \left( \frac{A_j}{A_d} \right)^2 \right] \right) \tag{19}$$

## 3. Fault Tolerant Control Strategy for Thrust Allocation

Upon the occurrence of the mentioned physical faults in the waterjet, it may be desired to alter the control allocation strategy in order to reduce the load on the faulty waterjet while maintaining the same path following performance. The waterjet load torque estimation scheme is shown in Figure 3 based on the diesel engine model and waterjet model introduced in Section 2. The reconfiguration of the control allocation is designed with the comparison between the actual waterjet power $\hat{P}_w$ and the desired one $P_{wd}$. The actual waterjet power is computed through an estimate of the waterjet load torque. When the discrepancy between $\hat{P}_w$ and $P_{wd}$ rapidly changes, a re-weighting of the cost terms in the

allocation part is performed and the reference heading speed for the path following is ready to be recomputed eventually.

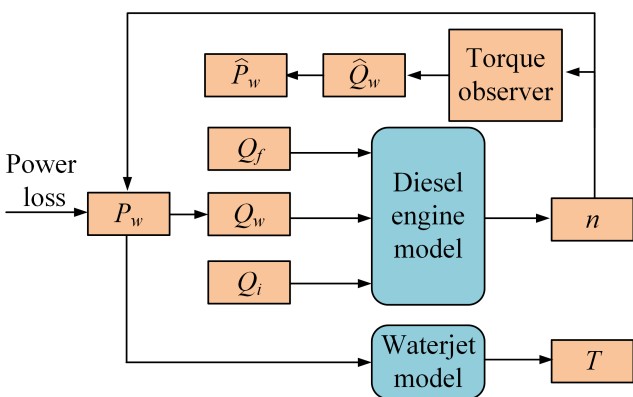

**Figure 3.** Waterjet load torque estimation scheme.

### 3.1. Design of the Load Torque Observer

Based on the engine dynamics presented in Section 2.1, the following dynamical model is adopted for the design of the state estimator of the waterjet load torque $Q_w$:

$$
\begin{aligned}
\dot{n} &= \theta_1 u + \theta_2 n + \theta_3 Q_w + \theta_4 \\
\dot{Q}_w &= -\frac{1}{T_Q} Q_w
\end{aligned}
\tag{20}
$$

where it is assumed that the waterjet torque $Q_w$ is a slowly time-varying quantity whose dynamical behaviour can be modeled as a first order low-pass filter with time constant $T_Q \gg 0$. To estimate the waterjet load torque, a Luenberger observer is adopted:

$$
\begin{aligned}
\dot{\hat{n}} &= \theta_1 u + \theta_2 \hat{n} + \theta_3 \hat{Q}_w + \theta_4 + L_1(y - \hat{y}) \\
\dot{\hat{Q}}_w &= -\frac{1}{T_Q} \hat{Q}_w + L_2(y - \hat{y})
\end{aligned}
\tag{21}
$$

where $L_1 > 0$ and $L_2 > 0$ are the observer gains and $y = n$ is the measured shaft rotational speed.

Let $\hat{\mathbf{x}} = [\hat{x}_1 \hat{x}_2]^{\mathrm{T}} = [\hat{n}, \hat{Q}_w]^{\mathrm{T}}$ be the state vector of the load torque observer. Considering the final form of the engine model in Equations (6) and (7), the state estimator is rewritten as:

$$
\begin{aligned}
\dot{\hat{x}}_1 &= \theta_1 u + \theta_2 \hat{x}_1 + \theta_3 \hat{x}_2 + \theta_4 + L_1(y - \hat{y}) \\
\dot{\hat{x}}_2 &= -\frac{1}{T_Q} \hat{x}_2 + L_2(y - \hat{y})
\end{aligned}
\tag{22}
$$

Since $\tilde{\mathbf{x}} = [\tilde{x}_1 = x_1 - \hat{x}_1, \tilde{x}_2 = x_2 - \hat{x}_2]^{\mathrm{T}}$ is the estimation error, the estimation error dynamics in vector-matrix form can be written as:

$$
\dot{\tilde{\mathbf{x}}} = \begin{bmatrix} \theta_2 - L_1 & \theta_3 \\ -L_2 & -\frac{1}{T_Q} \end{bmatrix} \tilde{\mathbf{x}} = \mathbf{A}\tilde{\mathbf{x}}
\tag{23}
$$

The stability of the origin of the estimation error dynamics is assessed by means of Lyapunov stability theory. Consider the positive definite candidate Lyapunov function

$V(\tilde{\mathbf{x}}) = \tilde{\mathbf{x}}^T \mathbf{P} \tilde{\mathbf{x}}$ where $\mathbf{P} = \frac{1}{2}\mathbf{I} > 0$ and $\mathbf{I}$ is the identity matrix; then the derivative of $V(\tilde{\mathbf{x}})$ along the trajectories of the systems is:

$$
\begin{aligned}
\dot{V}(\tilde{\mathbf{x}}) &= \frac{1}{2}\tilde{\mathbf{x}}^T(\mathbf{A} + \mathbf{A}^T)\tilde{\mathbf{x}} \\
&= \tilde{\mathbf{x}}^T \underbrace{\begin{bmatrix} \theta_2 - L_1 & \frac{1}{2}(\theta_3 - L_2) \\ \frac{1}{2}(\theta_3 - L_2) & -\frac{1}{T_Q} \end{bmatrix}}_{\mathbf{M}} \tilde{\mathbf{x}}
\end{aligned}
\tag{24}
$$

$\dot{V} < 0$ if the matrix $\mathbf{M}$ is Hurwitz, i.e., if $L_1 > \theta_2 - \frac{1}{T_Q}$. The observer gain $L_2$ determines if the eigenvalues of $\mathbf{M}$ (and thereby $\mathbf{A}$) are real or complex. In particular, if $0 < L_2 < L_2^\star$ then the eigenvalues of $\mathbf{A}$ are both real and negative, with $L_2^\star = \frac{1}{4\theta_3}(L_1 - \theta_2 + \frac{1}{T_Q})$. It follows that the origin of the estimation error dynamics is exponentially stable.

### 3.2. Reconfigurable Control Allocation

The control allocation scheme should account for the waterjet which is faulty by diminishing the demand in terms of delivered thrust. This is achieved by designing an adaptive thrust allocation algorithm that captures the knowledge of the power loss at each waterjet through adaptive weights. The proposed thrust allocation algorithm utilizes the estimated load torque $\hat{Q}_w$ to monitor the performance of the propulsion system and decide if reconfiguration is needed. In particular, the estimate $\hat{P}_w$ of the actual waterjet power is computed as

$$
\hat{P}_w = 2\pi n \hat{Q}_w \tag{25}
$$

which, in turn, is utilized to calculate the waterjet performance factor $\chi_p$:

$$
\chi_p = \frac{\hat{P}_w}{P_w} \tag{26}
$$

where $\chi_p \in (0, 1]$ and $P_w$ is the desired waterjet power obtained in Equation (16).

#### 3.2.1. Adaptive Weighting in Thrust Allocation

The adaptive weight $\omega$ is then defined as follows

$$
\begin{cases} \omega_f = \frac{1}{\chi_p}\omega_{f,0} \\ \omega_h = \chi_p \omega_{h,0} \end{cases} \tag{27}
$$

where $\omega_f$ is the weight for faulty waterjet and $\omega_h$ is the weight for the healthy one that is ready to compensate for the loss thrust command. $\omega_{f,0}$ and $\omega_{h,0}$ are the nominal weights.

Inspired by Chang et al. [12], the thrust allocation scheme consists of two consecutive steps. In the first step, the control vectors $X$, $Y$ and $N$ are distributed to the two waterjets' thrusts $T_{x_1}$, $T_{y_1}$, $T_{x_2}$ and $T_{y_2}$. Then, in the second step, the requested thrust is used to determined the control variables of each waterjet, i.e., engine speed, steering angle and reversing angle.

The objective function in the first stage of the optimization is defined as:

$$
\begin{aligned}
\min_{T_{xi}, T_{yi}} \quad & F_1(T_{xi}, T_{yi}) = \sum_{i=1}^{2} \omega_i\left[(T_{xi} - T_{xi0})^2 + (T_{yi} - T_{yi0})^2\right] + \mathbf{s}^T\mathbf{\Lambda}\mathbf{s} \\
\text{s.t.} \quad & T_{x1} + T_{x2} + s_x = X, \\
& T_{y1} + T_{y2} + s_y = Y, \\
& T_{x1} \times l_{x1} - T_{x2} \times l_{x2} - T_{y1} \times l_{y1} - T_{y2} \times l_{y2} + s_M = N, \\
& T_{y\,\min} \leq T_{yi} \leq T_{y\,\max}, \\
& T_{x\,\min} \leq T_{xi} \leq T_{x\,\max}
\end{aligned}
\tag{28}
$$

where $T_{xi0}$ is the initial value of longitudinal thrust, $T_{yi0}$ is the initial value of lateral thrust, $\mathbf{s} = [s_x, s_y, s_M]^T$ the vector of slack variables and $\mathbf{\Lambda}$ is the positive definite weighting matrix used to minimize $\mathbf{s}$. Further, $T_{x\,min}$ and $T_{x\,max}$ are the minimum and maximum value of the longitudinal thrust of each waterjet; $T_{y\,min}$ and $T_{y\,max}$ are the minimum and maximum value of the lateral thrust of each waterjet; $(l_{xi}, l_{yi})$ are the lateral and longitudinal distances of the waterjet from the vessel center of rotation.

For the second stage of the optimization, the objective function is given by:

$$
\min_{n_i, \alpha_i, \beta_i} \quad F_2(n, \alpha, \beta) = \sum_{2}^{i=1} \left(\frac{n_i - n_{i0}}{N}\right)^2 + \left(\frac{\alpha_i - \alpha_{i0}}{A}\right)^2 + \left(\frac{\beta_i - \beta_{i0}}{B}\right)^2 + I_n n_i^2
$$

$$
\begin{aligned}
\text{s.t.} \quad & T_{xi} = f_x(\mathrm{n}_i, \alpha_i, \beta_i), \\
& T_{yi} = f_y(\mathrm{n}_i, \alpha_i, \beta_i), \\
& n_{\min} \leq n_i \leq n_{\max}, \\
& \alpha_{\min} \leq \alpha_i \leq \alpha_{\max}, \\
& \beta_{\min} \leq \beta_i \leq \beta_{\max}
\end{aligned}
\tag{29}
$$

where $n_{i0}$ is the initial engine speed, $\alpha_{i0}$ is the initial steering angle, $\beta_{i0}$ is the initial reversing angle and $I_n$ is the weight used to minimize the power cost of the engine. Further, $N$, $A$, $B$ are the ranges of variation of the engine speed, steering angle and reverse angle. The functions $f_x$ and $f_y$ in Equation (29) describe the relationship between the control variables and the produced thrust in the longitudinal and lateral directions. Such functions were computed following the method presented in [12].

### 3.2.2. Heading Speed Reconfiguration

In some cases, the power loss may be so severe that the sole reallocation of thrust demands between the two waterjets will not suffice to deliver the necessary power to fulfill the desired heading speed. This ultimately will affect the capability of the vessel to maintain the desired course, hence leading to a poor path following performance. Thus, in addition to adjusting the weights in thrust allocation, the reconfiguration of the heading speed may be needed to achieve the desired mission within basic maneuverability of the vessel. The heading speed is therefore updated exploiting the performance factor $\chi_p$ according to the following

$$
U_d = \chi_p U_{d0}
\tag{30}
$$

The heading speed update is performed only when the integral of the heading error computed by the path following controller is larger than a user defined threshold, which indicates that the vessel may lose its turning ability totally in the operation.

## 4. Simulation Results

The fault-tolerant thrust allocation strategy is tested in simulation in two scenarios, zig-zag path following and sideways motion, where waterjet power loss faults of increasing severity are added. The line of sight guidance law [32] is used to compute the desired set-points of heading angle. A sliding model manoeuvring controller is then applied to compute the desired forces and moments to achieve the desired path following.

The vessel layout and the applied coordinate system are depicted in Figure 4. Table 2 gives the specifications of the twin-waterjet propelled vessel. The waterjet on the port side is referred to number '1' and that one on the starboard side is referred to number '2'. The ranges of variation of the engine speed, steering angle and reversing angle are $n \in [550 \text{ rpm}, 2450 \text{ rpm}]$, $\alpha \in [-40°, 40°]$, $\beta \in [0°, 45°]$, respectively. In the following simulations, the proposed reconfigurable thrust allocation strategy engages 10 s after the fault has happened to account for the fault diagnosis time period.

**Table 2.** Specifications of the twin-waterjet propelled vessel.

| Description | Symbol | Specification |
|---|---|---|
| Length between perpendiculars | $L$ | 10.8 m |
| Breadth | $B$ | 3.19 m |
| Draft | $D$ | 0.41 m |
| Block coefficient | $C_b$ | 0.42 |
| Tonnage of no-load state | $T_0$ | 7.8 t |
| Longitudinal separation between the waterjet outlet and the center of gravity | $L_G$ | 5.30 m |
| Lateral separation between the waterjet outlet and the center of gravity | $W_G$ | 0.64 m |

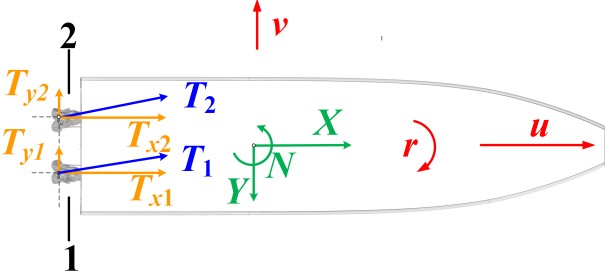

**Figure 4.** Waterjet layout and reference coordinate system.

### 4.1. Case 1: Zig-Zag Path Following

The zig-zag path to be followed is shown in Figure 5 with way-points given in the north-east inertial reference frame being $WP_1 = (0, 0)$, $WP_2 = (0, 350)$, $WP_3 = (-450, 600)$, and $WP_4 = (-450, 1400)$. As the surge velocity is much greater than the sway velocity, the initial heading speed is set as the desired surge velocity as 8 m/s. To show the effectiveness of the developed FT-CA strategy at different levels of waterjet load power loss, 10%, 30% and 70% power loss faults, corresponding to the $0.9P_w$, $0.7P_w$ and $0.3P_w$, are added to the waterjet '1' since 20 s, 90 s and 150 s, which are marked with green square boxes in the figure, respectively.

Figure 5 compares the overall simulation results with/without applying the heading speed reconfiguration strategy. The red line is the trajectory of the vessel and the shaded area is zoomed in to show the details of the vessel movement including the hull motion attitude. The two simulation results illustrate that the vessel may maintain steady and safe operation under the light power loss that happened at 20 s and 90 s, which means the adaptive thrust allocation suffices to compensate for the lost power. However, when the power loss hits 70% at 150 s into the simulation then the vessel no longer keeps the desired heading since the residual power does not suffice to meet both set-points in speed and heading. The heading speed reconfiguration strategy reduces the speed set-point, thereby allowing the vessel to retain the necessary manoeuvrability for path following. The heading speed reconfiguration is applied 10 s after the 70% power loss happened, which is shown in Figure 5b.

Figure 6 presents the outcome of the heading and speed control as the loss of power at waterjet '1' increases over time. Figure 7 gives the comparison between the desired waterjet load power $P_w$ and the estimated one $\hat{P}_w$, which is calculated using the proposed load torque observer. The blue dotted line is the performance factor $\chi_p$. It is shown that the estimator follows the desired signal before 30 s under fault-free condition and also it has the ability to track the change of load power when the power loss happened corresponding to the 10%, 30% and 70% power loss. The forces and moment $X$, $Y$ and $N$ as computed by the manoeuvring controller are depicted in Figure 8, whereas the corresponding thrust allocation outcome—$T_{x_1}$, $T_{x_2}$, $T_{y_1}$ and $T_{y_2}$—is presented in Figure 9. The figure shows that $T_{x_2}$ and $T_{y_2}$ obtain higher variation for the thrust reference based on the extent of the power loss, which clearly indicates that the adaptive thrust allocation shifts the thrust demand from the faulty to the healthy waterjet. The engine speed, steering angle and reversing angle resulting from the changes in thrust demands are depicted in Figure 10.

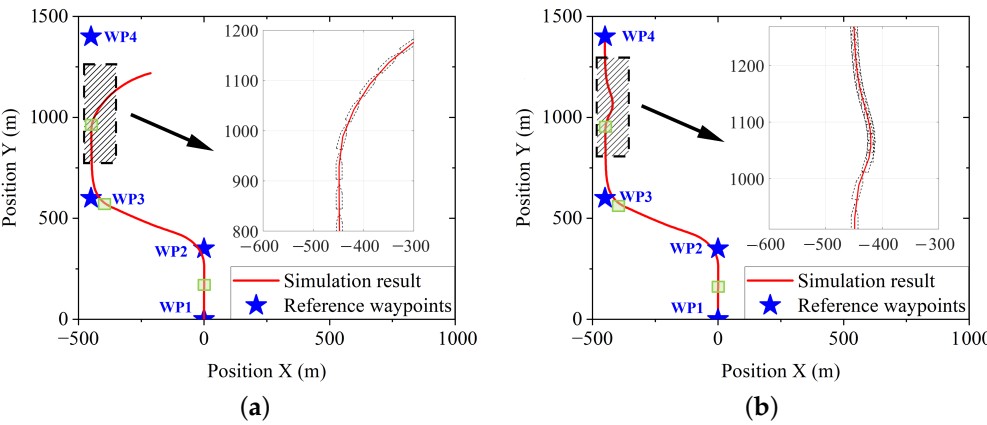

**Figure 5.** Zig-zag path following results. (**a**) Without surge velocity reconfiguration; (**b**) With surge velocity reconfiguration.

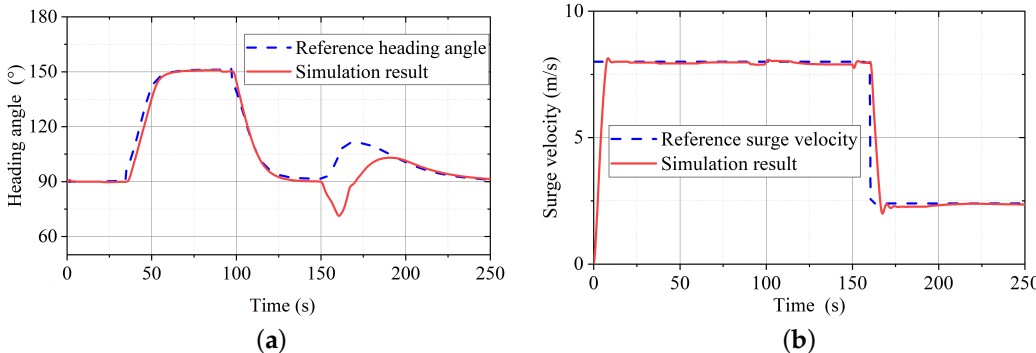

**Figure 6.** Measurements from navigation system in zig-zag path following. (**a**) Heading angle following; (**b**) Heading speed following.

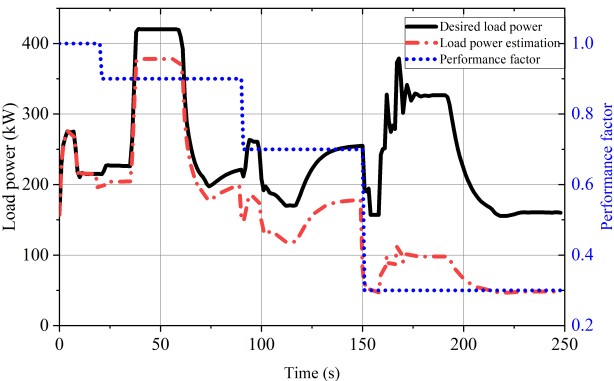

**Figure 7.** Waterjet load power estimation result.

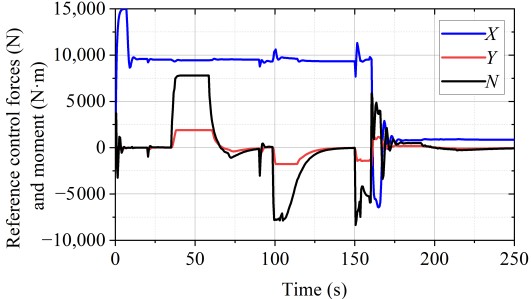

**Figure 8.** Reference control vectors in the zig-zag path following.

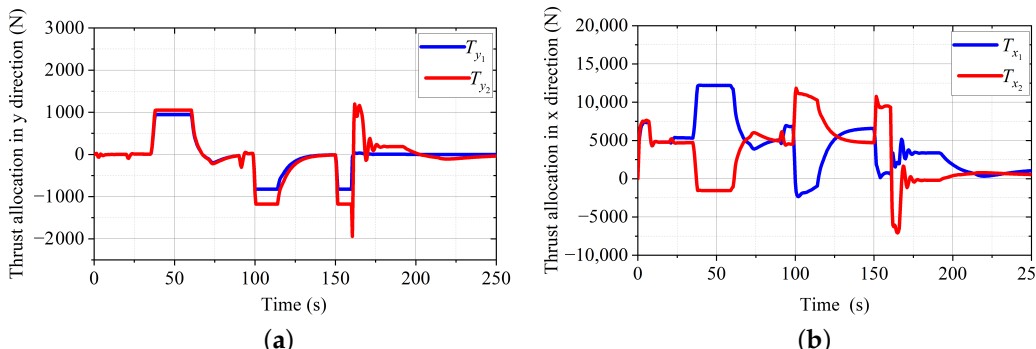

**Figure 9.** Thrust allocation results of reference control signals. (**a**) Thrust allocation in y direction; (**b**) Thrust allocation in x direction.

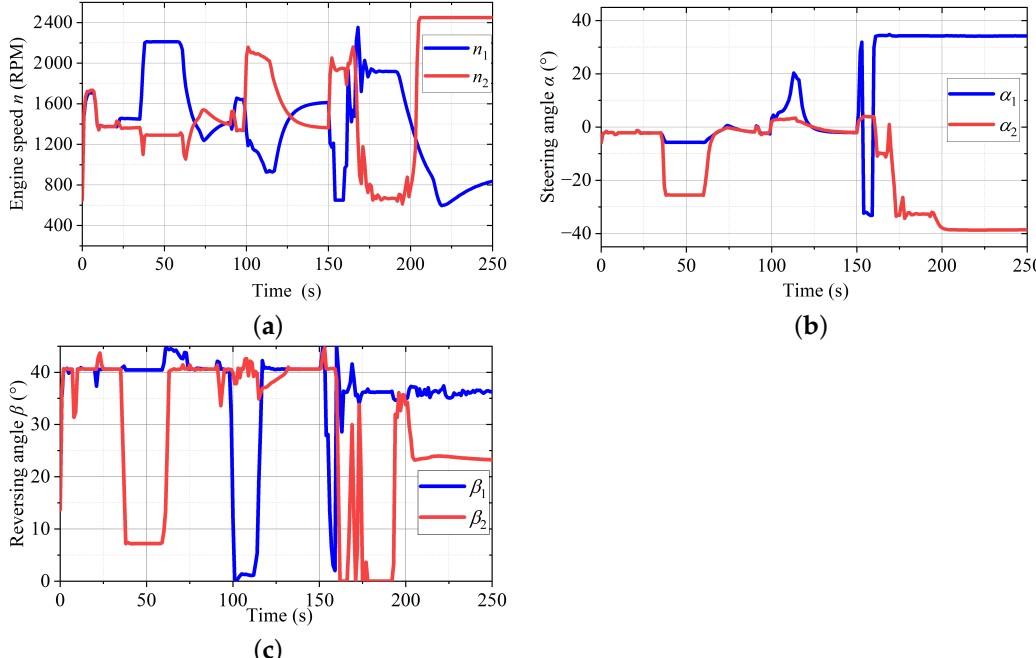

**Figure 10.** Allocation results of waterjet actuators. (**a**) Engine speed allocation results; (**b**) Steering angle allocation results; (**c**) Reversing angle allocation results.

### 4.2. Case 2: Sideways Movement

Sideways movement is a common and useful way for waterjet vessel docking, which is considered to be critical and complex since the vessel operates in a constrained area where tight motion control is required [33,34]. In sideways movement simulation, the vessel moves laterally toward its port side while it maintains the desired heading at the same time. In this case the sway velocity is considered the desired heading speed. The reference start point and end point are $WP_1 = (0, 0)$ and $WP_2 = (0, 150)$, respectively. The desired sway velocity is set to 1 m/s and the desired surge velocity and heading angle are both set to zero. Power loss faults of 10%, 30% and 70%, corresponding to $0.9P_w$, $0.7P_w$ and $0.3P_w$, are added to waterjet '1' since 30 s, 50 s and 90 s, respectively.

Similarly to what is shown for the zig-zag manoeuvre, the heading speed reconfiguration strongly improves the overall control performance of the vessel during the sideways motion when the power loss is at 70% as shown in Figure 11. Figure 12 presents the heading angle, the surge velocity and sway velocity related to the sideways movement simulation. For the mild faults, like 10% and 30% power loss, the waterjet vessel easily overcomes the diminished power at waterjet '1' by adjusting the demands for waterjet '2'.

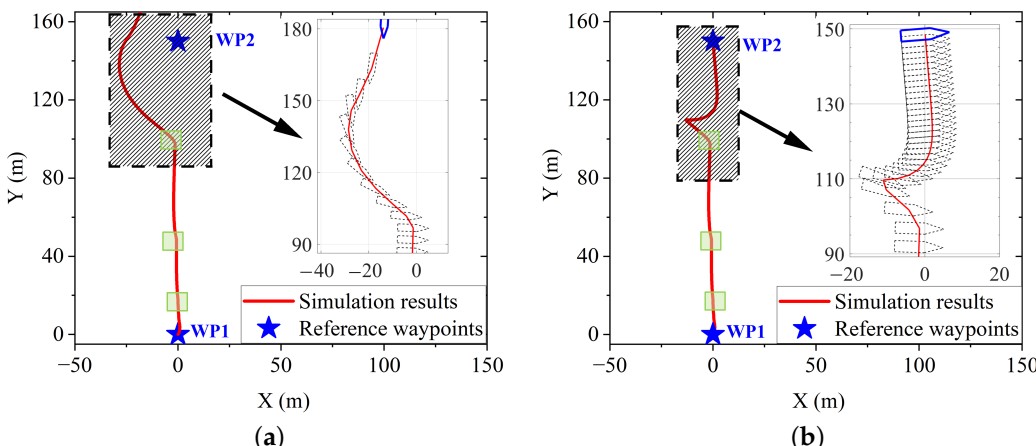

**Figure 11.** Sideways movement results. (**a**) Without sway velocity reconfiguration; (**b**) With sway velocity reconfiguration.

Figure 13 shows the comparison between the desired waterjet load power and the estimated one during sideways movement in the presence of increasing power loss. The reference control vector delivered to the vessel is shown in Figure 14 and the corresponding thrust allocation results are presented in Figure 15. In sideways movement, the thrust component in the latter direction of both waterjets is the main force driving the lateral motion of the vessel. Also in this simulation scenario, the adaptive thrust allocation strategy responds to the increasing power loss on waterjet '1' by increasing the thrust demand on waterjet '2', as illustrated in Figure 15a.

The results of engine speed, steering angle and reversing angle calculation corresponding to the thrust allocation results are shown in Figure 16. The significant operation after the severe power loss happened at 110 s further confirms the very good manoeuvring possibility of the waterjet vessel in such a fault scenario.

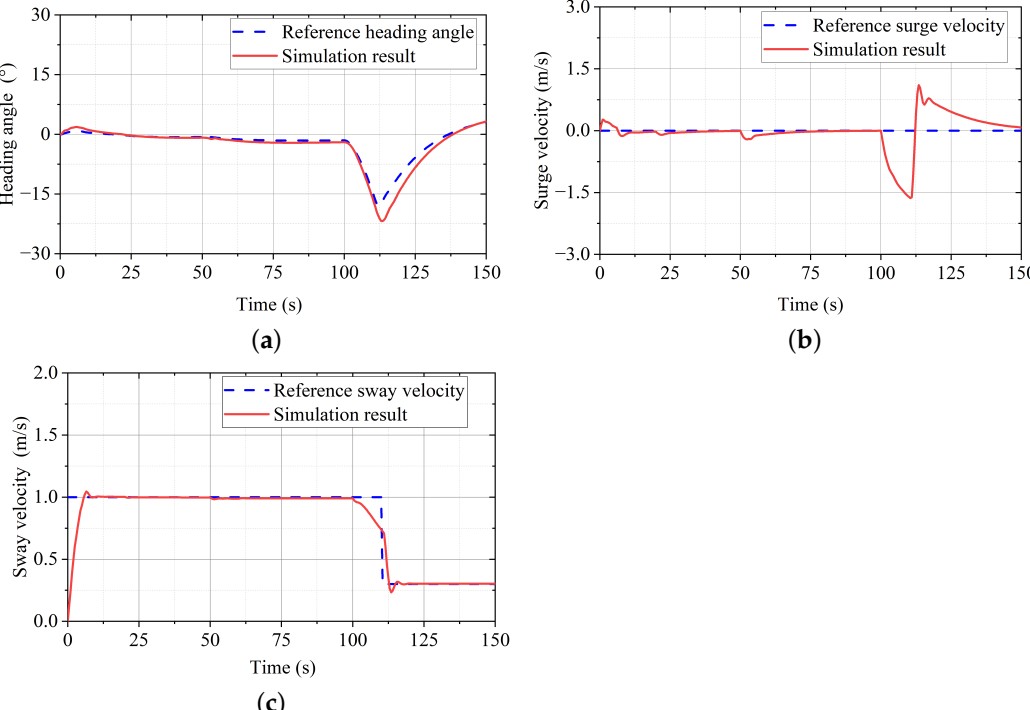

**Figure 12.** Measurements from navigation system in sideways movement. (**a**) Heading angle following; (**b**) Surge velocity following; (**c**) Sway velocity following.

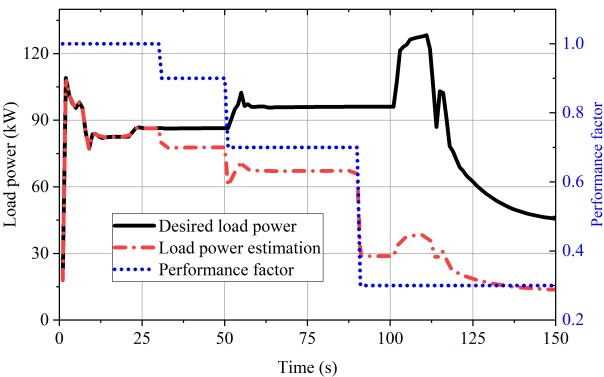

**Figure 13.** Waterjet load power estimation result.

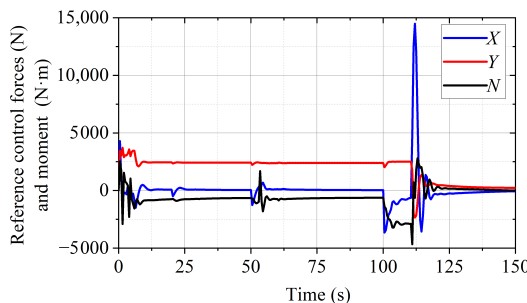

**Figure 14.** Reference control vectors in the sideway movement.

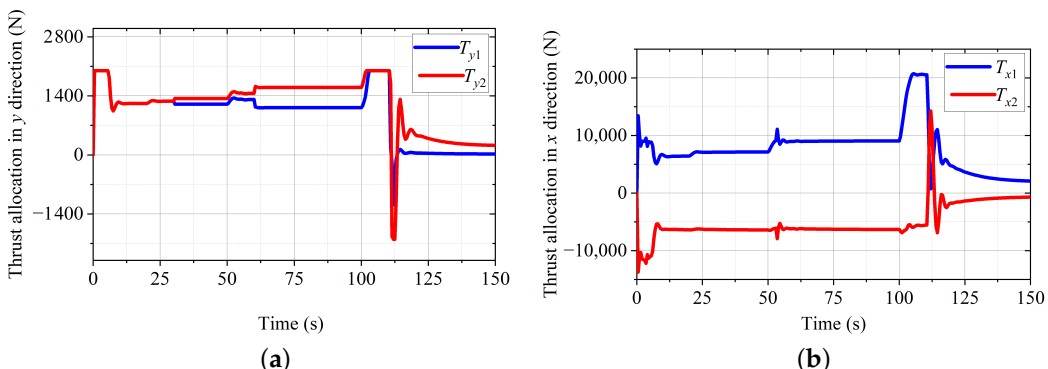

**Figure 15.** Thrust allocation results of control signals. (**a**) Thrust allocation in y direction; (**b**) Thrust allocation in x direction.

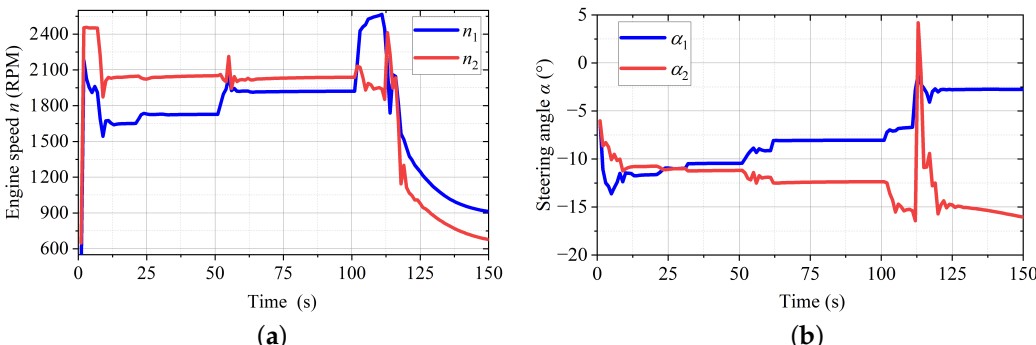

**Figure 16.** *Cont.*

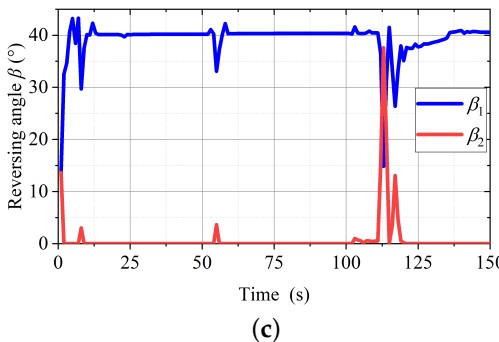

**(c)**

**Figure 16.** Allocation results of waterjets actuators in the sideway movement. (**a**) Engine speed allocation results; (**b**) Steering angle allocation results; (**c**) Reversing angle allocation results.

## 5. Conclusions

This paper presents a fault tolerant control allocation strategy for the twin-waterjet propelled vessel. The thruster dynamics is addressed in the research and a model of the the engine and the waterjet is presented. Power loss faults are considered within the scope of research. The proposed fault-tolerant scheme integrates a load-torque observer and a reconfigurable thrust allocation optimization algorithm. The load torque observer serves to track changes in actual waterjet power and enables the computation of the waterjet performance factor, which ultimately triggers the thrust allocation reconfiguration. Adaptive weights are then introduced in the performance index of the thrust allocation strategy to push towards a higher utilization of the healthy waterjet. In addition, the paper proposes a complementary heading speed reconfiguration to reduce the manoeuvring performance in the face of severe power losses at one of the waterjets. Simulation results for zig-zag and sideways motion showed that, in the presence of increasing power loss, the weights in the thrust allocation as well as the setpoints of heading speed are reconfigured to shift the thrust demands from the faulty to the healthy waterjet. In real operations, such a strategy could prevent the fault to quickly escalate into a complete failure of the waterjet due to excessive power demand.

**Author Contributions:** Conceptualization, Z.X. and R.G.; writing—original draft preparation, Z.X.; writing—review and editing, Z.X. and R.G.; supervision, R.G. and J.Y. All authors have read and agreed to the published version of the manuscript.

**Funding:** This research was funded by China Scholarship Council (202006230222).

**Institutional Review Board Statement:** Not applicable.

**Informed Consent Statement:** Not applicable.

**Data Availability Statement:** Not applicable.

**Conflicts of Interest:** The authors declare no conflict of interest.

## Nomenclature

The following abbreviations are used in this manuscript:

| | |
|---|---|
| FTC | Fault Tolerant Control |
| TA | Thrust Allocation |
| $h_s$ | total loss, m |
| $h_{in}$ | inlet loss, m |
| $h_{out}$ | outlet loss, m |
| $h_d$ | duct loss, m |
| $h_h$ | head loss, m |
| $q_{mf}$ | fuel injection flow, kg/s |
| $v_j$ | nozzle outlet velocity, m/s |
| $v_s$ | inflow velocity, m/s |

| | |
|---|---|
| $A_d$ | duct area, m$^2$ |
| $A_j$ | area of nozzle outlet, m$^2$ |
| $C_m$ | piston's mean speed, m/s |
| $H_L$ | constant of caloric value, MJ/kg |
| $J$ | rotational inertia, kg $\cdot$ m$^2$ |
| $P_f$ | average friction force, kW |
| $P_w$ | nominal waterjet power, kW |
| $\hat{P}_w$ | waterjet load power estimation, kW |
| $Q_i$ | indicated torque, N·m |
| $Q_f$ | friction moment, N·m |
| $Q_w$ | waterjet load torque, N·m |
| $\hat{Q}_w$ | waterjet load torque estimation, N·m |
| $T_x$ | longitudinal thrust, N |
| $T_y$ | lateral thrust, N |
| $U_d$ | desired heading speed, m/s |
| $V$ | diesel cylinder volume, L |
| $\alpha$ | utilization coefficient |
| $\beta$ | kinetic energy utilization coefficient |
| $\eta_i$ | thermal efficiency |
| $\omega_f$ | weighting for faulty waterjet |
| $\omega_h$ | weighting for healthy waterjet |
| $\rho$ | density of water, kg/m$^3$ |
| $\chi_p$ | performance factor |

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
