# Peer review of "Fault-Tolerant Thrust Allocation with Thruster Dynamics for a Twin-Waterjet Propelled Vessel"

_jmse, doi:10.3390/jmse10121983_

Round 1
Reviewer 1 Report
The authors in this paper developed a fault tolerant control allocation strategy for the twin-waterjets propelled vessel. However, I have some considerations that if they are answered, it can improve the manuscript:
1. In the paper twin-waterjet propelled vessel in the study, but absent information about type of twin-waterjet vessel and technical parameters. Must be added information about the types and technical parameters of twin-waterjet for developed a fault tolerant control.
2. In the section 3.1 authors wrote about stability, but in the simulation part absent result of analysis of stability.
3. In the lines 85-86 authors wrote that the occurrence of severe cavitation power efficiency may decrease. Must be added estimation or citation about impact of cavitation on power efficiency.
4. Must be added estimation of the power efficiency by developed a fault tolerant control, also extended a discussion of results.
With best regards,
Author Response
Dear Reviewer,
Thank you for taking your time to review this manuscript. On behalf of all the contributing authors, I would like to express our sincere appreciations of all your comments and suggestions. Please find the itemized responses and revisions/corrections in the re-submitted files.
Best regards,
Zijing Xu

Reviewer 2 Report
Expand the literature survey related to the twin-waterjet propelled vessel.
The novelty and superiority of this investigation should be stated clearly at the end of the literature surveys.
Add clearly descriptions for all equations, and add abbreviations also nomenclature.
It's suggested to add more variables such as the load torque.
Add dynamic analysis related to the waterjet flow.
Author Response
Dear Reviewer,
Thank you for taking your time to review this manuscript. On behalf of all the contributing authors, I would like to express our sincere appreciations of all your comments and suggestions. Please find itemized responses and revisions/corrections in the re-submitted files.
Best regards,
Zijing Xu

Reviewer 3 Report
Notes in the attachment - the article is acceptable after minor corrections.

Author Response

(The authors gave the same response as above.)

Reviewer 4 Report
The authors present a fault tolerant control strategy for a vessel equipped with two orientable waterjet thrusters.
The authors model the actuator dynamics in detail, they design a linear Luenberger observer to estimate the fault, and they adjust the weights in the control allocation scheme to accommodate for the fault.
The objective is clear and the novelty is fair.
The following technical issues should be addressed in my opinion.
In Section 2, please declare the measurement unit of each variable for the sake of clarity: some coefficients are clearly related to the conversion of measurement units. A table may be useful for the purpose.
In Section 2.1, I am not fully convinced by the modeling in (6).
In my experience, the input-output equations for a thruster are non-linear, while (6) seems to be linear.
In your case of a diesel engine, for example:
- you lump $C_m^2$ (see equation 5) into the constant parameter $\theta_4$, but I feel $C_m$ linearly depends on the engine speed $n$, isn't it? So a $n^2$-like nonlinearity is neglected in (6).
- how do you calculate $Q_w$ in your simulation? Is it something like $P_w/n$ (see $P_w$ in equation 14)? If this is the case, again, $Q_w$ depends $n^2$, according to (15).
I feel your approach can be practically acceptable, despite the nonlinearities, by performing a first order approximation.
If you do so (i.e., equation 6 represents a linear approximation of a nonlinear model) please specify it, otherwise clarify the model.
Please check if I'm right and, if necessary, explain why such nonlinearities can be neglected.
In Section 2.2, equation (14), you state that the fault may consist in $h_s$ increasing.
In the simulation results, you refer to a fixed "power loss" (i.e., a percentage) instead.
Did you simulate the fault increasing $h_s$ in the simulation results?
Or did you change the power efficiency $\eta_p$ in (14) instead?
I have found no plots nor explanations, so I suggest to clarify it.
In Section 3.1, after equation (25), I cannot understand why you talk about input-to-state stability: you have already shown that the error is globally exponentially stable, as expected, so why do you introduce ISS?
In Section 3.2, I noticed that $X_p$ in (27) is just a single scalar value.
Do you assume only one waterjet is faulty?
So, is $X_p$ calculated only for the faulty waterjet (if any, $X_p=1$ otherwise)?
In Section 3.2.1, I believe you should include an additional term in the cost function in (29).
You have 4 independent variables and 3 equality constraints, so you have some redundancy.
As far as I understood, $T_{xi0}$ and $T_{yi0}$ represent the thrusts in the previous sampling time (the current thrust).
If so, your cost function is designed to minimize the rate of the optimization variables, but you do not minimize the total thrust: this may lead to inefficient orientation of the thrusters (e.g, one facing the other).
A possible solution is to add a term such as $+\gamma(\omega_1 T_{x1}^2 + \omega_2 T_{x2}^2 + \omega_1 T_{y1}^2 + \omega_2 T_{y2}^2)$ in the cost function, that is equivalent to the weighted squared-2-norm of the two thrusts, where $\gamma>0$ is a scale factor to balance the requirements. The problem is still a QP, so the complexity is almost the same, but I feel such approach is more generic and solid.
In Section 3.2.1, please specify how the problem (30) is solved.
Problem (29) is a small quadratic programming problem, so plenty of solvers are available, while (30) introduces nonlinear constraints. Do you employ sequential quadratic programming or maybe an interior point algorithm to solve (30)? Is the computational effort feasible in view of a real online implementation?
In Section 4, I wonder why no plots for the torque observer ($\hat{P}_w$, $\hat{Q}_w$) and for the estimation of $X_p$ are proposed.
Moreover, remark 1 is a bit confusing: you have designed an observer (that intrinsically introduces a delay in the estimation), so why do you need to add a fixed delay to mimic the fault diagnosis delay?
Do you actually implement your observer in simulation to estimate $X_p$? Do you introduce some measurement noise to test how the observer behaves in a more realistic scenario?
Finally, I have found some typos and grammar should be revised. Please proofread the simulation results section in particular.
I have done my best to report the typos I have found in order of appearance:
- no necessary to
- Figure 2, The
- torque$Q_w$ (no space)
- is calculate as (many times)
- the nominal waterjet power $P_e$ ($P_w$ maybe?)
- the discrepancy ... get rapid change
- and $T_{y,2}$) (extra bracket)
- may lost
- is worked $10$ second after
- which shown the well dynamic performance of the motion controller
- It is illustrate (many times)
- could more flexible to compensate
- delivered to the are
Author Response

(The authors gave the same response as above.)

Round 2
Reviewer 1 Report
Paper was corrected in line with review.
It can be accepted in the present form.
Reviewer 4 Report
I thank the authors for considering this reviewer's suggestions.
I have no further questions, just a note: I have found a typo in the abstract, i.e., "This papers addresses" instead of "This paper addresses".